# Inspiring Anti-Tick Vaccine Research, Development and Deployment in Tropical Africa for the Control of Cattle Ticks: Review and Insights

**DOI:** 10.3390/vaccines11010099

**Published:** 2022-12-31

**Authors:** Paul D. Kasaija, Marinela Contreras, Halid Kirunda, Ann Nanteza, Fredrick Kabi, Swidiq Mugerwa, José de la Fuente

**Affiliations:** 1SaBio, Instituto de Investigación en Recursos Cinegéticos (IREC), Consejo Superior de Investigaciones Científicas (CSIC), Universidad de Castilla-La Mancha (UCLM)-Junta de Comunidades de Castilla-La Mancha (JCCM), Ronda de Toledo s/n, 13005 Ciudad Real, Spain; 2National Livestock Resources Research Institute (NaLIRRI), Kampala P.O. Box 5704, Uganda; 3National Agricultural Research Organisation (NARO), Mbarara ZARDI, Mbarara P.O. Box 389, Uganda; 4College of Veterinary Medicine, Animal Resources and Biosecurity, Makerere University, Kampala P.O. Box 7062, Uganda; 5Department of Veterinary Pathobiology, Center for Veterinary Health Sciences, Oklahoma State University, Stillwater, OK 74078, USA

**Keywords:** ticks, anti-tick vaccines, anti-tick antigens, vaccinomics, tick-borne diseases, reverse vaccinology, personalized vaccines, immune response

## Abstract

**Simple Summary:**

Ticks affect the health of humans and animals. In tropical African countries, cattle are threatened by ticks and the diseases they transmit, making necessary a tick control to avoid endangering the production of this industry. In this review, different vaccine candidates and different approaches are proposed as an environmentally friendly control alternative that can be included in the strategies of these countries to prevent tick-borne diseases their consequences.

**Abstract:**

Ticks are worldwide ectoparasites to humans and animals, and are associated with numerous health and economic effects. Threatening over 80% of the global cattle population, tick and tick-borne diseases (TTBDs) particularly constrain livestock production in the East, Central and Southern Africa. This, therefore, makes their control critical to the sustainability of the animal industry in the region. Since ticks are developing resistance against acaricides, anti-tick vaccines (ATVs) have been proposed as an environmentally friendly control alternative. Whereas they have been used in Latin America and Australia to reduce tick populations, pathogenic infections and number of acaricide treatments, commercially registered ATVs have not been adopted in tropical Africa for tick control. This is majorly due to their limited protection against economically important tick species of Africa and lack of research. Recent advances in various omics technologies and reverse vaccinology have enabled the identification of many candidate anti-tick antigens (ATAs), and are likely to usher in the next generation of vaccines, for which Africa should prepare to embrace. Herein, we highlight some scientific principles and approaches that have been used to identify ATAs, outline characteristics of a desirable ATA for vaccine design and propose the need for African governments to investment in ATV research to develop vaccines relevant to local tick species (personalized vaccines). We have also discussed the prospect of incorporating anti-tick vaccines into the integrated TTBDs control strategies in the sub-Saharan Africa, citing the case of Uganda.

## 1. Introduction

Ticks are obligate hematophagous arthropod ectoparasites distributed worldwide [1], and belong to two families; Ixodidae (hard-bodied ticks) and Argasidae (soft-bodied ticks). They affect 80% of the world’s cattle population and are associated with numerous health and economic effects [2]. In developing tropical countries, tick-borne diseases (TBDs) constitute a major constraint to the livestock production, especially among smallholder farms of East, Central and Southern Africa [3,4,5]. They inflict direct damage to their host and are ranked only second to mosquitoes as vectors of disease [6]. The economic impact of ticks is strongly linked to the epidemiology of the diseases they transmit, and is expressed as both direct and indirect losses [2]. Thus, in sub-Saharan Africa, TBDs are considered the most important animal diseases [7,8].

The enormous loss associated with TTBDs could be minimized by controlling tick populations to acceptable levels [9]. Currently, tick control is majorly based on the use of chemical acaricides [10]. However, the improper and continuous use of these chemicals has increased the incidence of acaricide-resistant ticks [11], and occurrence of environmental and food contamination [12,13]. Consequently, there is increasing economic and social demand for alternative and advanced tick control methods [9] that can mitigate the negative effects of acaricides. The non-chemical control methods currently applied include: use of genetically resistant animals [14], pasture management strategies, environmental modifications and biological control technologies [15].

Livestock farmers in tropical African countries such as Uganda are spending a significant amount of money on tick control using acaricides, which is performed at least once a week [16,17,18,19]. The methods used for acaricide application are crude, labor-intensive and time-consuming, and hence, ineffective. The acaricides are very expensive and there are many brands on the market (up to 25 brands). These are also adulterated at various levels. Overuse and misuse of acaricides has further compromised the effectiveness of the chemicals and led to the selection of drug-resistant ticks. Therefore, despite routine application of acaricides, the challenge of TTBDs still persists [17,20,21,22].

Anti-tick vaccines (ATVs) are considered a favorable alternative to chemical tick control, since they reduce food and environmental contamination, are cost-effective, prevent pathogen transmission, can reduce acaricide resistance and can potentially be applied in various hosts [23,24,25]. The mechanism of protection by ATVs is based on the development of antigen-specific antibodies that interact with and affect the function of the targeted antigen in the tick feeding on an immunized host [6,26]. Thus, ATVs can affect tick feeding and reproduction, controlling tick infestations by reducing tick weight, oviposition and fertility, which, in turn, indirectly reduces pathogen prevalence [1,27].

The Gavac^plus^ and TickGard ATVs, which are based on *Rhipicephalus microplus* antigens, are the only commercially registered anti-tick vaccines, and have been used in Cuba, Mexico, Venezuela and Australia against the cattle ticks *R. microplus*, *R. australis* and *R. annulatus* with varying levels of efficacy that range between 10 and 89% [27,28]. The vaccines generally reduced tick populations, the number of acaricide treatments and babesiosis and anaplasmosis infections in Cuba and Colombia, respectively [27]. Although their sustained use was associated with various technical and commercialization challenges, proof of concept of anti-ectoparasite vaccines has inspired the discovery of many potential tick protective antigens, some of which can be fast-tracked for new vaccine formulations [27].

Given the TTBD control situation in sub-Saharan African countries, intervention by ATVs would be a useful undertaking. Unfortunately, there has been limited research and no deployment of ATVs in these countries [29]. Moreover, putative anti-tick antigens (ATAs) have mainly been identified in and tested against ticks of foreign origin to Africa, which may render them irrelevant to some African tick species [30]. This may partly explain why the BM86/95-based vaccines showed limited protection against diverse tick species of foreign geographical origins, showing a 46% reduction in the number of engorging *B. decoloratus* adult females and 56% and 61% reductions in cumulative tick and egg weights, respectively [31,32].

In order to improve protection against multi-tick species, development of multi-antigen or chimeric vaccines that incorporate critical tick and pathogen-derived antigens [30,33,34] have been proposed to reduce both tick infestations and pathogen infection/transmission [35,36]. Currently, however, the prospects of developing such a universal anti-tick vaccine are low. Therefore, it is imperative to design tailor-made or personalized vaccines for tick populations of specific geographical locations [37].

This review, therefore, advocates for the boosting of anti-tick vaccine research, development and deployment in sub-Saharan Africa, and presents some of the logical considerations that guide researchers to identify vaccine antigens. We summarize the characteristics of a desirable anti-tick antigen and outline how biotechnology tools can be used to identify and modify an antigen to achieve higher efficacy. We finally give our opinion of the prospect of incorporating anti-tick vaccines into the integrated TTBD control program in sub-Saharan Africa.

## 2. Origin of Anti-Tick Vaccines and Immunological Control of Tick Infestations

Control of tick parasitism using immunological strategies has been studied for more than 40 years [38,39]. Earlier studies have revealed that some bovines acquired an immunological response to ticks from the inoculation of various antigens and molecules of tick salivary gland origin [40,41]. These observations form the basis for ATV research and development [9].

Anti-tick vaccines became commercially available in the early 1990s to control tick infestations in cattle [42,43]. Despite how long they have existed, alternative and more effective ATVs are still not available. Identification of molecules essential for both tick survival and for host–vector–pathogen interactions have been hypothesized as strategies for the development of novel vaccines, and for simultaneous control of ticks and tick-borne pathogens [36,44]. In the recent years, a number of tick proteins (candidate antigens) have been identified and assessed in controlled pen trials, yielding variable results [30,45] (Appendix A).

Elvin and Kemp, 1994 [46] proposed a candidate antigen for the development of an anti-tick vaccine as one for which the host antibodies can sufficiently gain access to the target protein. The formation of the antibody–antigen complex can disrupt the function of the target protein and/or induce physiological changes that affect tick biology, and conserved epitopes are shared among several tick species to protect against multiple vector infestations [1] (Appendix A). In addition, a combination of proteins expressed in different tissues or at different stages of the vector’s life cycle could produce more potent vaccines, since different stages and tissues could be targeted [47].

## 3. The Bm86/95-Based Anti-Tick Vaccines

Early in the 1990s, the Bm86/Bm95-based vaccines became the first and only anti-tick vaccines ever approved and registered for commercial use. TickGARD (in Australia) and Gavac (in Latin American countries) are both derived from *R. microplus* midgut membrane-bound recombinant protein Bm86. Bm86 is a glycoprotein present in the cell membrane, and is involved in endocytosis [48]. Cattle vaccination with Bm86-based vaccines shows a positive correlation between antigen-specific antibodies and reduction in tick infestations and fertility (i.e., vaccine efficacy). This observation permits the evaluation of vaccine efficacy through the measurement of anti-Bm86 antibody titers in vaccinated animals [12]. It has, however, been observed that the physiological status of the animals (age and reproductive status) affects the primary antibody response to vaccination, but not subsequent vaccination responses [42].

The protective effect of the vaccine is achieved when anti-Bm86 antibodies interact with the Bm86 protein molecules to interfere with the biological function of the protein, resulting in reduction in tick numbers, weight and reproductive capacity of engorging female ticks [42]. Vaccination trials with Gavac resulted in reduction in tick infestations as well as the incidence of babesiosis and *Anaplasma phagocytophilum* infections among vaccinated cattle herds [42,49,50].

Although Bm86-based vaccines are generally effective for controlling tick infestations in cattle, field trials showed variation in the degree of protection, ranging between 41% and 100% against different tick strains in different cattle breeds, and the vaccines were mainly effective against the *Rhipicephalus* species [51]. De Vos et al. (2001) [52] observed no significant protection by the Bm86 antigen against *Rhipicephalus appendiculatus* and *Amblyomma variegatum* ticks. The difference in the susceptibility of the ticks can be attributed to physiological differences between tick species, and those encoded in the sequence of Bm86 orthologs [53].

Although vaccines have not ultimately solved the tick problem in countries where they have been used, they are undoubtedly suitable targets for research aimed at tick control [9]. There is an increasing number of new promising vaccine targets that can affect both tick infestations and pathogen transmission. These are aimed at addressing the limitations of the Bm86-based vaccines.

## 4. Exploring Tick Biology for Antigen Identification and Vaccine Development

Knowledge of specific physiological processes of ticks has been utilized to rationally develop promising vaccine candidates that can impair tick biological processes. These targeted physiological processes effect tick attachment to the host, uninterrupted feeding on the host, intracellular digestion of large blood volumes and metabolism of ingested blood into massive clutches of eggs laid by the engorged female ticks [54]. Some of the biological processes that have been studied to identify target antigens for designing candidate anti-tick vaccines are discussed below, as well as in Appendix A.

**(a)** 
**Tick Attachment and Feeding to Repletion**


Interfering with tick attachment to the host would be the ideal intervention for preventing both tick feeding and pathogen transmission to the host [54]. During tick attachment, an Ixodid tick secretes glycine-rich proteinaceous cement-like substances in the saliva that harden around the inserted mouthparts [55]. This cement cone enables the tick to remain attached to the host during the long duration of feeding (up to 14 days) and prevents host immune mediators from accessing the tick’s proboscis. Thus, a vaccine targeting components of the cement cone could ideally interfere with tick attachment and pathogen transmission [40].

The tick bite stimulates host defenses such as itch, homeostasis, inflammation and immune response. Homeostasis (blood coagulation, vasoconstriction and platelet aggregation), complement activation and inflammation constitute the early protection against tick infestation [56]. The processes leading to acute inflammatory response begin when host tissue is first damaged, but it is the subsequent migration and degranulation of white blood cells, particularly the granulocytes, at the bite site that mark the beginning of inflammation. This may occur within three hours and can last several hours. While pathogens such as the Powassan virus require a short transmission time (15 min) and may elude the inflammatory response, most bacterial and protozoan pathogens require several hours of tick feeding before transmission [43,57]. It has been demonstrated that the cellular response attracting inflammatory cells to the feeding site of *Phlebotomus papatasi* is sufficient to block transmission of Leishmania [32]. Upon tick attachment and feeding, both cellular and humoral mediators of vertebrate adaptive immunity are activated, with T and B memory cells amplifying the host inflammatory response to subsequent tick infestation through cytokine and antibody production [56,58].

Uninterrupted feeding, therefore, requires ticks to counter the complex host immune responses mounted against them. Tick saliva is a complex mixture of bioactive molecules that are used by the tick to modulate, deviate or inhibit various cellular and molecular functions of the vertebrate’s defense mechanisms, creating an immune-privileged environment at the bite site, which facilitates transmission of pathogens to the vertebrate host [56,59]. This phenomenon is termed saliva-assisted transmission (SAT) [60]. Ticks have evolutionarily developed a range of molecules that counteract almost all the vertebrate’s immune defenses. Gene expression and production of saliva molecules is upregulated soon after tick attachment. Correlation of real-time tick salivary gland transcript and protein expression with corresponding changes in host skin and regional lymph node gene expression elucidates the complex interaction between the tick and host responses [59,61,62], which can guide identification of anti-tick antigens for vaccine development. Although some saliva molecules have been identified and their functions described [59,60], there is extensive redundancy at the molecular, cellular and functional level, and the proteins generally exhibit low immunogenicity [56,59].

A 29-kDa salivary protein (p29) was identified by Mulenga et al. (1999) [63] while screening the cDNA library of *Haemophysalis longicornis* with rabbit immune serum raised against tick saliva proteins. Due to its structural homology to collagen, the protein was presumed to be a constituent of the extra-cellular matrix that forms the cement cone during tick attachment. When recombinant p29 was used to immunize rabbits against *H, longicornis* reduced female tick engorgement weights and caused mortalities of 40–55% among larvae and nymphs.

Salivary proteins HL34 and HL35 were identified by Tsuda et al. (2001) [64], by immune-screening of a cDNA library of an adult *H. longicornis* combined with amplification and cloning of the genes. Since expression of the HL34 and HL35 genes is induced during the slow feeding phase, both in the tick salivary glands and in other organs, the proteins are suspected to play a role in tick feeding. The presence of proline and tyrosine repeat amino acid domains, which characterize adhesive molecules, also suggest the proteins to be components of the cement cone. Vaccination of rabbits with recombinant HL34 protein affected nymphs and reduced oviposition in adults due to impaired blood digestion.

Another cement cone component protein (36kDa) designated *Rhipicephalus* Immuno-dominant Molecule 36 (*RIM36*) was identified from the *R. appendiculatus* cDNA library. The protein is principally localized in the *e* cells of the type III salivary gland acinus, in which *Theileria parva* sporozoites also develop. During tick feeding, *RIM36* induces a strong host antibody response, to which some studies partly attribute *R. appendiculatus* resistance among experimental guinea pigs [65]. Recombinant *RIM36* also reacted with immune sera from cattle either experimentally infested with ticks or obtained from field infestations [40].

Tick protein 64P, identified in *R. appendiculatus,* is a putative secreted component of the cement cone. Its compositional similarity to the host skin proteins suggests evolutionary mimicry to avoid rejection of the tick by the host’s immune response. For better exposure of epitopes to the host’s immune system, recombinant truncated constructs of 64P (64TRPs) were fused with GST and used to immunize rabbits. These elicited both humoral and cell-mediated immune responses among the vaccinated animals, which was amplified following tick infestation. This amnestic immune response and the observed local cutaneous inflammatory response are prerequisites to development of naturally acquired resistance to tick infestation, and are desirable as a candidate for an anti-tick vaccine. Cross-reactivity of anti-64TRPs antibodies with salivary, midgut and hemolymph epitopes of both adult and nymph *R. appendiculatus* subsequently caused mortalities after tick detachment. Similar protective effects of *R. appendiculatus* 64TRPs were observed against adult and nymphal *Rhipicephallus sanguineus* and *Ixodes ricinus* [66], indicating the potential of 64TRPs as a broad-spectrum anti-tick vaccine. In addition, *R. appendiculatus* 64TRPs successfully protected mice against the tick-borne encephalitis virus (TBEV) transmitted by infected *I. ricinus* ticks to a level comparable to that of a dose of commercial TBEV vaccine [67].

**(b)** 
**Immunomodulation and regulation of enzyme activity**


Mammalian hosts can acquire immunity (resistance) against ticks as a result of prolonged infestation or vaccination with tick antigenic proteins, which affects tick physiological processes such as feeding, reproduction and viability [58]. As noted in (a) above, a tick bite induces host homeostatic and immune regulatory responses, which interfere with the tick’s attachment and feeding [68,69]. A variety of proteases, notably serine proteases, are involved in the mediation of mammalian homeostatic and immune processes. The activity of these enzymes is regulated by a group of proteins collectively known as protease inhibitors [70,71].

In arthropods, serine protease inhibitors (serpins) are presumed to regulate endogenous homeostatic processes and to protect against infection by inhibiting pathogen-derived proteases. Similarly, ticks are likely to deploy serpins to counter host homeostatic and immune responses to facilitate uninterrupted feeding or to maintain their own physiology. This, therefore, makes serpins a potential target as an anti-tick candidate antigen [72]. Serpins are produced in different tick organs, such as the salivary glands [73], the gut [74] and the hemolymph [72], and different tick species target different serine proteases. For example, the serpin HLS2 has only been demonstrated in the tick *H. longicornis,* and regulates tick endogenous proteases during feeding. A candidate recombinant anti-tick vaccine based on HLS2 (r HLS2) yielded about 40% mortality for tick nymphs and adults fed on vaccinated rabbits. It was, however, not suitable for use as a single vaccine antigen [72].

Other serpins which have been evaluated as possible anti-tick vaccines include *Rhipicephallus appendiculatus-*derived RAS-1 to -4 [75], among which a combination of rRAS-1 and -2 reduced nymph engorgement by 60% and increased adult tick mortality by 28–43% in cattle immunization trials [76]. Other combinations of these serpins mostly affected parasite-infected ticks.

Cystatins constitute a different superfamily of protease inhibitors targeting papain-like cysteine proteases and legumains, and are present in vertebrates and invertebrates. In soft and hard ticks, type 1 (also called stefins) and type 2 cystatins have been identified, and these only share limited amino acid sequence similarity with cystatins in other organisms (<40%). They have been mostly described in tick salivary glands and the midgut, where they may possibly affect blood digestion. The Bmcystatin in *R. microplus* midgut plays a role in the tick’s embryogenesis, since it inhibits the vitellin-degrading cysteine endopeptidase (VTDCE) [77]. Similar to Bmcystatin is the midgut stefin, Hlcyst-1 of *H. longicornis*, which regulates host blood digestion by inhibiting the hemoglobinolytic activity of a cathepsin L-like cysteine protease, HlCPL-A. Type-2 cystatins are secretory in nature and have been studied extensively. Among these are sialostatins L (SL) and L2 (SL2) from *I. scapularis,* which are capable of inhibiting cathepsin L, with SL additionally inhibiting cathepsin S. It is noteworthy that cathepsins S, L and V are vital for mammalian immunity due to their involvement in antigen presentation processes of dendritic cells and macrophages [78].

Experiments silencing sialostatins L and L2 prevented tick feeding on rabbits by 40%, demonstrating their role in tick blood feeding. Sialostatins L2 was particularly upregulated in salivary glands of feeding ticks, and when used as a recombinant vaccine in animal experiments, early rejection of ticks at feeding sites or prolonged feeding times were observed. A strong immunosuppressive effect of sialostatins in the host was also observed [79,80]. Other type 2 cystatins described in *H. longicornis* include Hlcyst-2 and -3 (midgut) and HLSC-1 (salivary glands). These are capable of inhibiting papain and cathepsin L, with Hlcyst-2 additionally inhibiting Cathepsin H [81,82]. Hlcyst -2 also interacts with HlCPL-A to affect host blood digestion in a way similar to Hlcyst-1 [83,84]. Hlcyst -2 is additionally implicated in the tick’s innate immunity, as shown by in vitro experiments with *Babesia bovis* [82]. Among the soft ticks, type 2 cystatins identified in *O. moubata* include om-cystatin 1 and 2. These strongly inhibit papain and cathepsin B and H, but om-cyastatin 2 also binds cathepsins C, L and S [85,86]. Due to the immunosuppressive action of tick salivary components, such as cystatins, their neutralization by host antibodies or gene silencing in ticks can significantly reduce tick feeding and, possibly, control tick infestation [83].

**(c)** 
**Osmoregulation (water balance)**


Unlike insects, which use malpighian tubules, hard ticks rely on salivary glands for osmoregulation [87]. In order to attain full engorgement, an ixodid female tick can suck 200–300 times its own weight in blood. Using the salivary glands, the tick actively excretes up to 70% of the excessive fluid and ions back into the feeding lesion, to concentrate blood components for efficient digestion and to allow further intake of blood [87,88]. Part of this large salivary flow constitutes the transmission route for pathogens and bioactive molecules which modulate the host’s immunity [89]. On the other hand, during long spells when an unfed tick is off the host, salivary glands can produce a hyperosmotic secretion which facilitates the absorption of atmospheric moisture [87]. Given the critical importance of osmoregulation and water balance in tick physiology and survival, water channels (aquaporins) constitute a suitable target for designing an anti-tick vaccine. Aquaporins (AQPs) are protein structures that render the lipid bilayer of cell membranes permeable to water [90], and their role in the salivary glands of *I. ricinus* has been demonstrated [87]. Three aquaporins have been identified in cattle tick *R. microplus,* and are designated RmAQP1, RmAQP2 and RmAQP3. Cattle vaccination with recombinant RmAQP1 yielded an efficacy ranging between 68–75% [91].

**(d)** 
**Blood digestion (Hemoglobinolysis)**


Besides survival, the major importance of a blood meal for an adult female tick is maintenance of the reproductive vigor, which is measured by the massive egg production [9]. Tick blood digestion occurs intracellularly in the gut epithelial cells, where it is carried out by a network of acidic peptidases. These hemoglobinolytic enzymes mainly comprise aspartic endopeptidases (cathepsin D-like), cysteine endo- and exopeptidases (cathepsin L, B and C type) of the papain family, asparaginyl endopeptidases (legumain peptidases) and monopeptidases [92]. These hemoglobinases have also been functionally characterized in various tick species, mainly based on gene-specific RNAi silencing [93,94], and their gene expression has been shown to be induced and upregulated by tick feeding [84,94,95]. Since their molar concentration and activity increases with feeding, with most of them peaking at full engorgement, a vaccine targeting hemoglobinolytic enzymes may not block pathogen transmission. Moreover, whereas vaccination using recombinant antigens of these enzymes stimulates high antibody titers, only limited efficacy was detected, possibly due to the high redundancy of their coding genes [96].

**(e)** 
**Heme and iron transport and storage**


Whereas other blood-sucking arthropods excrete excess heme and iron through feces or polymerize it into insoluble hemozoin, the hemoglobin ingested by ticks is phagocytosed by specialized epithelial cells of the midgut, and is digested intracellularly inside large acidic vesicles [97,98]. A network of acidic lysosomal peptidases is involved in the hydrolytic process [99], generating great amounts of heme inside the cells. Consequently, ticks always have to contend with excessive amounts of toxic heme and iron-derived metabolites from their blood meal [100]. Heme is capable of catalyzing the formation of reactive oxygen radicals, which can, in turn, cause oxidative damage and disrupt the cellular lipid bilayer [101].

As an adaptation to heme toxicity, ticks have evolved heme detoxification mechanisms. During the first days following a blood meal, heme is mainly absorbed from the midgut and transferred into the hemocoel, where it is bound by hemolymph hemelipoprotein (HeLp), which transports and delivers it to peripheral tissues, particularly the ovaries. Since HeLp forms the bulk of hemolymph proteins, heme noticeably accumulates into the oocytes and ovaries (making 80% of ovarian proteins), which facilitates vitellogenesis [100,102,103]. In ticks, vitellin, which is the main yolk protein, is a hemoprotein [104]. Towards the end of blood digestion, most of the heme generated (>90%) aggregates into specialized organelles called hemosomes, and only a limited amount is used by the tick’s own metabolic demands [100,102]. Heme in the tick’s body is always bound to a protein to prevent toxicity [100], and for HeLp, the lipoproteic component serves as an antioxidant [105]. In addition, various peptide products of hemoglobin digestion perform an antimicrobial role (hemocidins), augmenting ticks’ immunity [106]. Due to the critical role played by heme-binding and trafficking proteins (detoxification), targeting them could enable the development of new tick control strategies [100].

Although iron is essential for various biochemical processes, such as oxygen transport, energy metabolism and DNA synthesis, it can catalyze the generation of reactive oxygen species, which can cause oxidative damage to cells and tissues [107]. Different organisms, therefore, use various proteins for the safe metabolism of iron. The protein ferritin, present in most organisms, is important for iron metabolism. Depending on the animal species, it is involved in iron storage, homeostasis, protection against oxidative damage and iron transport in insects [108]. In ticks, ferritin is important for blood feeding and reproduction, and two ferritins have been characterized. Intracellular ferritin 1 (FER1), produced in midgut cells, stores iron within the cell, while a secretory ferritin, FER 2, transports iron to peripheral tissues such as the ovaries and oocytes. Ferritin 2 is, therefore, important for oviposition and embryonic development. Silencing ferritin-coding genes reduces feeding, body weight and fecundity while increasing mortality and morphological defects [108].

Rabbit immunization studies with recombinant ferritins have shown that the proteins are immunogenic and that they induce the production of high levels of antibodies, with anti-FER 2 antibodies being present even in eggs. Since FER 2 is abundant in hemolymphs and circulates in the tick’s body, ingested host antibodies gain significant access to it. The protein (FER2) also exists exclusively in ticks. Thus, with a vaccine efficacy of approximately 50% against *H. longicornis* in rabbits [109], and over 60% and 70% against *R. microplus* and *R. annulatus,* respectively, in cattle vaccination trials [24], it is believed that FER 2 (recombinant), if formulated in combination with other antigens, may improve the efficacy and cross-protectivity of either antigen [109].

**(f)** 
**Detoxification (elimination of toxic substances)**


Glutathione S-transferases (GSTs) are a family of enzymes present in various tissues of eukaryotic organisms, and are involved in the metabolic detoxification of xenobiotics, reactive oxygen species, heme and other endogenous compounds [110]. In the presence of glutathione (GSH), GSTs’ catalytic reactions result in less harmful products, which are easily excreted by the cell [111]. The increased activity of GSTs in some pesticide- and drug-resistant parasite strains compared to susceptible ones demonstrates the detoxification function of the enzymes [112,113]. In addition, several acaricides have been shown to target GSTs activity, supporting the possible use of GSTs as a candidate anti-tick antigen [114,115], and since the antigen is relatively conserved across tick species, the induced immune response may be cross-protective [110]. Cattle vaccination trials with *H. longicornis* recombinant GST antigens yielded a 50% vaccine efficacy [54]. Similar studies by Parizi et al. (2011) [110] recommended the use of GSTs in combination with other characterized antigens to boost efficacy.

**(g)** 
**Embryogenesis (yolk accumulation and degradation)**


The tick population in the environment is partly maintained by the tick’s capacity to lay large volumes of eggs that give rise to greater numbers of offspring per subsequent generation. It is, therefore, logical to control tick populations by interfering with their reproductive processes, such as vitellogenesis, embryogenesis and fertility. It is also considered possible to target internal tissues, since studies have shown host antibodies to circulate in the hemolymphs of ticks feeding on immunized hosts [116]. Vitellogenesis or yolk accumulation, which is a process during which extraovarian and ovarian tissues produce protein precursors that are conveyed to and selectively accumulate in the oocytes [117,118], is critical to the tick’s reproductive success. The major protein sequestered from the hemolymph by developing oocytes is vitellogenin, which crystallizes to be stored as vitellin [117] in structures called yolk spheres or yolk granules. Vitellin forms the source of amino acids for tick embryonic development [119]. It is, therefore, imperative to understand the composition of yolk proteins, the tissues where they are produced, how they are transported and sequester in the oocytes, and how they are enzymatically mobilized during embryogenesis. Known among these are two Aspartic endopeptidases, *Boophilus* yolk pro-cathepsin (BYC) and Tick heme-binding aspartic peptidase (*THAP*), as well as Cathepsin-L-like vitellin degrading cysteine endopeptidase (VTDCE).

Produced in the gut and fat body, BYC is secreted into the hemolymph and sequesters in the oocytes [120], constituting up to 6% of egg protein [121]. The enzyme is involved in the hydrolysis of vitellin in tick eggs [120] and possibly hemoglobinolysis in the larvae [122]. Cattle immunization with native and recombinant BYC stimulated specific IgG responses with protective efficacies of up to 36% and 25%, respectively, the vaccine effects being notable in the number and weight of engorged females, as well as in egg fertility [123]. Tick heme-binding aspartic peptidase (THAP) is a VT-degrading enzyme present in tick eggs, whose activity can be inhibited by heme at a site remote to the catalytic site [119]. Thus, by binding heme, THAP plays a role in maintenance of the redox balance, preventing oxidative damage [124]. Vitellin-degrading cysteine endopeptidase (VTDCE), is cathepsin-L-like, which refers to a group of enzymes present in all tick developmental stages. Exhibiting higher vitellin-hydrolytic activity than BYC and THAP, VTDCE is distributed in the gut, ovary and hemolymph [118,125,126]. Animal vaccination trials with native VTDCE yielded a vaccine efficacy rate of 21%, the major effects being on egg weights and number of engorging females [125]. The *Boophilus microplus* Cathepsin-L 1 (BmCL1-recombinant) or RmLCE (native form) is a cysteine endopaptidase found in tick larvae where it hydrolyses hemoglobin and VT subunits generated by previous activity of maternal enzymes (VTDCE, BYC and THAP).

**(h)** 
**Enzymatic disruption and remodeling of host tissues**


Naturally, metalloproteases (MPs) are multipurpose proteins involved in many biological functions, and are present in various organisms [23]. In ticks, salivary MPs are used in the remodeling or disruption of host tissues’ structural constituents, as well as interfering with homeostasis [127,128,129]. They are implicated in the degradation of fibrin and fibrinogen at the bite site [130], the inhibition of microvascular endothelial regeneration and the breakdown of cell integrins [60,131]. These enzymatic activities collectively impair natural wound healing at the tick bite site [132], and facilitate pathogen transmission to the host, e.g., Borrelia spirochetes [130].

Gene expression of metalloproteases is induced by tick feeding. A vaccine targeting these proteases could, thus, result in early tick rejection. Using a cDNA library, Decrem et al. (2008) [133] identified and designated two homologous MPs as Metis 1 and 2 in the tick *I. ricinus*. Characterization of their function by RNAi revealed reduced capacity of salivary gland extracts (SGEs) to disrupt fibrinolysis, while gene knock-down by the same method caused incomplete engorgement and mortalities. Recombinant Metis 1 reduced feeding and oviposition among immunized rabbits, but caused no effect on tick nymphs. Further, three other blood meal-induced MPs were described, and these showed limited genetic similarity to the former [132,133].

The tick *H. longicornis* metalloprotease (HLMP1) was demonstrated in salivary glands of all instars, and when its recombinant form (rHLMP1) was used to vaccinate rabbits, mortalities of 15.6% and 14.6% in nymphs and adults, respectively, were observed [134]. On the other hand, a 60% protection rate was obtained when recombinant MPs from *R. microplus* were used in bovine immunization trials. The vaccine mostly affected tick numbers, oviposition and egg hatching [23]. However, these and other studies show that although a vaccine based on MPs can advantageously produce an amnestic immune response, it does not sufficiently suppress tick infestation. This could possibly be due to the extreme redundancy of the antigen as envisaged by presence of MPs in various tissues, and its occurrence as multiple isoenzymes [133,134].

**(i)** 
**Tick engorgement and development of reproductive structures**


For most Ixodid ticks, there is a transition weight between the slow and rapid phases of tick feeding, termed the critical weight. When this is not attained by a female tick before detachment, no eggs can be laid. On the other hand, most unmated female ticks do not feed beyond critical weight no matter how long they remain on the host. It is believed that a testicular engorgement factor (EF) is introduced into the feeding female during copulation, and stimulates rapid engorgement. This protein was identified from a cDNA library of fed *A. haebraeum* testicular tissue, and was designated voraxin [135]. It is a combination of two synergistically bioactive peptides, designated rAhEFα and AhEFβ, whose production is upregulated during feeding. The protein also induces salivary gland degeneration and partial development of the tick ovaries. It is believed that a vaccine based on voraxin would reduce oocyte development and pathogen transmission between the tick and the host. This is supported by the observation that the voraxin-based candidate vaccine yielded a reduction in the mean weight of 72% among surviving ticks, compared to 37% by BM86 based vaccines. Other physiologically vital mating factors considered in ticks include the sperm capacitation factor, which stimulates the final phase of sperm cell maturation within the female tick after copulation, and the Vitellogenesis-stimulating factor of the soft tick *O. moubata*, which is important for oviposition [136].

## 5. Insights into the Future

Although vaccines based on recombinant Bm86 and its orthologues have been successfully used in Latin America and Australia against *R. microplus*, with efficacies ranging from 51 to 91% [30], vaccine trials show insufficient protection against other tick species and a limited effect on pathogen transmission [6]. Besides sequence homology, the failure of Bm86 to protect against a heterologous tick challenge was attributed to a number of other factors [28,32,53,137]. While many other antigens have been identified over time by different research groups, they still fall short of the desired high efficacy and cross-protectivity [138]. There is, therefore, a need to identify new tick and pathogen-derived antigens to improve anti-tick vaccine efficacy and cross-protectivity [54]. Furthermore, since some tick species parasitize different vertebrate hosts and share habitats and hosts with other tick species, the development of vaccines effective in different hosts and against several tick species is an emerging area of research interest [139].

### 5.1. Reverse Vaccinology

During recent years, reverse vaccinology has been used extensively to identify suitable antigens for the development of anti-tick vaccines. Reverse vaccinology involves rational selection of suitable tick samples and generation and sequencing of their transcriptomes to identify candidate anti-tick antigens, followed by evaluation of their antigenicity and possible adverse effects in the host [140]. It is dependent on a battery of sophisticated bioinformatics methods that analyze genomic data to identify and characterize antigens [141] (Figure 1). Such data may include B-cell epitopes and signatures of putative antigens, among others. For further evaluation of the identified antigens, reverse genetics can be employed to manipulate the target genes, followed by determination of the resulting phenotype [54]. A number of methods can be used to achieve this, including random or insertional mutagenesis and homologous recombination, but tick research has mostly employed RNA interference (RNAi) to characterize gene function. This suffers the challenge that the tick genome bears a large number of genes, most of whose functions are neither known nor predicted. However, the current availability of the *I. scapularis* genome/proteome serves as a reference for gene studies in other tick species [54].

### 5.2. Vaccinomics and Systems Biology

In order to develop more efficient vaccines, a systems biology approach is required for the holistic study of molecular mediators and pathways of the host–vector–pathogen interaction. It is believed that these molecular interactions are essential for tick survival and pathogen transmission, and that co-evolution of the tick, host and pathogen has a genetic bearing. The application of last-generation omics technologies to tick vaccine research will provide effective screening platforms and algorithms for the discovery of new tick-protective antigens [1]. Advances in genomics, transcriptomics and proteomics are facilitating global characterization of tick proteins for inclusion in a universal anti-vector vaccine. Vaccinomics is a modern research approach that integrates omics technologies and bioinformatics, and can be used in a systems biology study to identify and fully characterize candidate-protective antigens for the development of next-generation vaccines [142].

Defined as the integration of immunogenetics and immunogenomics with systems biology and immune profiling to understand the basis of immune response to vaccination [143,144], vaccinomics can elucidate outcomes of vaccination and guide the design of better vaccines (Figure 1). Immunogenetics focuses on individual host genetic variation associated with individual differences in immune responses to the same antigen(s), while immunogenomics considers population-level genetic variations associated with population-level variations in immune response [145]. Thus, vaccinomics can facilitate in-depth understanding of bovine immune responses to various antigens, so that vaccine design and formulation can be optimized to yield higher efficacy (Figure 1). Vaccinomics could also be useful in selecting suitable tick and pathogen antigen combinations for the design of next-generation vaccines (Figure 2, Appendix A).

### 5.3. Quantum Vaccinomics

Quantum vaccinomics focuses on the actual physical or functional units that are involved in the immune reactions to produce the observed outcome. Considering light, where a photon is defined as the quantum (smallest unit) of light, immune-protective epitopes were proposed to be the immunological quantum [146]. Quantum vaccinomics have been applied in the study of host–vector–pathogen molecular interactions to identify the protective epitopes or immunological quanta constituting the interacting protein domains. These can be used to design multi-epitope-based antigens (chimeric antigens) with various immunological advantages [142,147] (Figure 2).

## 6. Anti-Tick Vaccines in Africa

Uganda: Exotic cattle breeds (majorly dairy) and their crosses are more susceptible to TTBDs than indigenous breeds, and require heightened protection. Adoption of these commercial breeds in large areas of southwestern and central Uganda demands the maintenance of tick-free herds. The animals are mostly kept under a paddocking and small holder dairy production system (zero-grazing), which makes tick control by vaccination possible. On the contrary, indigenous cattle in many parts of sub-Saharan Africa are mainly managed extensively under pastoral and agro-pastoral systems. The animals are openly grazed and move several kilometers in search of community grazing pastures, which are available in rangelands. These rangelands may also be grazed by wild herbivores such as buffaloes, zebras, rhinos and antelopes, which also harbor ticks and provide a source to the livestock. Fortunately, these cattle are relatively resistant to tick infestations and can tolerate ticks for longer periods without intervention. Most veterinarians suggest limited tick control for these local breeds [7].

Tick Subolesin (SUB) is a transcription factor involved in the regulation of cellular processes, including feeding and the innate immune response to pathogen infection [148]. The protective effect of the antigen was evaluated against ticks infecting local and cross-breed cattle in Uganda. Recombinant SUB proteins from the three most economically important tick species in the country (*R. appendiculatus*, *R. decoloratus* and *A. variegatum*) showed over 75% amino acid similarity, and when evaluated in cattle clinical trials, there was a negative correlation between antibody titers and tick development. However, simultaneous vaccination with all three SUB antigens above shows limited protection against tick infestation, while *R. appendiculatus* SUB was more cross-protective. Considering the results of all tick species, the three tick developmental stages and the cattle breeds, an overall vaccine efficacy of 65% was achieved without significant differences (*p* = 0.82) between SUB antigens [149].

In another study, the orthologue of gut protein Bm86 was identified in *R. appendiculatus* (designated Ra86) with two variants—Ra85A and Ra92A. When recombinant Ra86, expressed in the baculovirus-insect cell system, was evaluated in rabbit immunization trials against larval, nymphal and adult *R. appendiculatus*, a tick mortality of 23.1% was observed among the adult ticks compared to 1.9% in controls. Both the mean weight of engorged female ticks and egg production reduced by 31.5% in rabbits vaccinated with the Ra86 recombinant protein compared with the controls. However, the vaccine did not affect the larval or nymphal stages of tick development [150]. Presently, both SUB and Ra86 are being studied further by separate research groups in Uganda for the possible development of candidate vaccines relevant to ticks in this geographical region [149,150].

Kenya: The Bm86-based vaccine-TickGARD™ Plus was evaluated against *R. appendiculatus* among Bos indicus calves. The vaccine produced limited protection against the ticks, but caused a significant reduction in the mean engorged weight of *B. decoloratus* and the egg weight per surviving adult female tick. The homologues of Bm86 from *R. appendiculatus* (designated Ra86) obtained from a laboratory tick stock and from four Kenyan field populations revealed varying degrees of amino acid polymorphisms, while phylogenetic analysis showed inter-specific variation. Recombinant Ra86 also induced an insignificant protective effect against adult female *R. appendiculatus* in rabbits. It was, thus, concluded that, while the TickGARD™ Plus vaccine had no potential to controlling adult *R. appendiculatus*, the Ra86 antigen could be used in a vaccine against *R. appendiculatus,* or in combination with other Bm86 homologues or *Theileria parva*-derived vaccines to control both ectoparasite and, possibly, pathogen transmission [151].

In another investigation evaluating TickGARD against *R. decoloratus*, a high amino acid sequence identity (85–86%) was observed between the Bd86 and Bm86 homologues. Native Bd86 in *B. decoloratus* and recombinant Bd86 strongly reacted with sera from TickGARD-vaccinated cattle, and we were able to identify two linear peptides conserved between the Bd86 homologues and Bm86 in an epitope mapping study. These results imply a possible ability of the vaccine to cross-protect against heterologous tick species with multiple antigen sequences [32]

Nigeria: The Bm86 gene homologues from the ticks *Hyalomma truncatum, Rhipicephalus annulatus* and *Rhipicephalus decoloratus* were characterized for the possible development of an anti-tick vaccine. A 100% nucleotide identity was observed amongst the *Rhipicephalus* species, but the sequence was divergent from that of *H. truncatum*. The phylogenetic analysis revealed a 3–8% sequence variation between the host and nucleotide sequences from the USA, Australia, Israel and South Africa, which may imply limited cross-protection [152].

Tunisa: Molecular characterization of Bm86 orthologues of *Hyalomma* ticks was carried out. Analysis of partial sequences of the Bm86 gene and its orthologues from *Hyalomma* tick species revealed increasing diversity rates, ranging from 0.26 to 6.02%, between *H. excavatum, H. anatolicum, H. marginatum* and *H. scupense* ticks. Amino acid sequence comparison between Bm86 and *H. excavatum* orthologues He86-A1/A2/A3 revealed high diversity (33–34%), which could decrease the efficacy of vaccination by commercial and experimental vaccines based on Bm86. However, there was limited (10.2%) amino-acid diversity between Hd86-A1 used in an experimental vaccine against *H. scupense* and He86-A1/A2/A3, suggesting that the Hd86-A1 vaccine candidate might be more appropriate to target the *H. excavatum* tick than the corresponding Bm86 vaccines [153].

By and large, limited research and published information exists on anti-tick vaccines in sub-Saharan African countries, and this is the motivation for this paper: to enlighten, inspire and mobilize African scientists to consider anti-tick vaccines as a possible integral tick control method in the wake of escalating acaricide resistance, especially among small-holder dairy farms keeping high-grade animals.

## 7. Opinion on the Possible Impact of Incorporating Anti-Tick Vaccines into the Integrated Approach for Tick Control: Case of Uganda

Deployment of effective ATVs in Uganda will help farmers to partly overcome the challenges associated with acaricides, resulting in decreased cattle mortality and morbidity caused by TTBDs. This will lead to increased profitability both for farmers and the country’s economy, since the livestock industry significantly contributes to the GDP [154]. Strategic integration of vaccination with acaricide application relative to seasonal tick populations [155,156] can be adopted for cost-effective and environmentally friendly control of tick infestation [25] in Uganda. Vaccinated animal populations will develop carrier status to TBDs, and the antibody titers will be higher (enzootic stability), changing the epidemiology of TTBDs. Herd immunity against TTBDs will develop after subsequent vaccinations, with a corresponding reduction in TTBD incidence [157].

Learning from the experience of anti-tick vaccines in Australia and Latin America, where vaccination programs were cut short because of technical and commercialization challenges, it is prudent to plan for the effective and sustainable deployment of ATVs. It was observed that ATVs were most successfully used in state-sponsored integrated tick control programs (e.g., in Cuba), which facilitated proper vaccine use and implementation [27]. Therefore, in Uganda, the government should invest in ATV research, development and local production, specifically focusing on the resident tick species. This can reduce costs and ensure continuous supply of the ATV doses for research and farmer consumption. Furthermore, the government can make ATVs part of established immunization programs, such as the FMD and Lumpy Skin disease vaccination. The different stakeholders (farmers, traders, local administrative bodies, etc.) should be sensitized and consulted at the different stages of vaccine development [19]. The government and the private sector can supply and monitor vaccine performance in the field. The cost effectiveness of integrating the various TTBD control strategies should be assessed, and policy makers should be advised [158].

## 8. Conclusions

Tick infestation continues to be a serious global impediment to livestock production. Effective and cost-effective integrated tick control strategies will combine acaricides with ATVs and other tick control methods to reduce the use of chemicals and its associated challenges. Strategic alteration of acaricides and ATVs, considering seasonal tick populations, can control TTBDs and boost farmer incomes.

Advances in omics technologies and their application in modern research approaches, such as reverse vaccinology, systems biology, vaccinomics and quantum vaccinomics, are revolutionizing the discovery of new anti-tick antigen targets. Consequently, the number of proteins with value as antigens has rapidly increased in recent years, and their assessment in vaccination trials has yielded promising results. Among these are AQP1, FER 2, Asparaginyl endopeptidases and cathepsins B, D and L. With the completion of more tick gene projects and continued use of reverse vaccinology, more efficacious anti-tick antigens can be identified and characterized. Current technology enables mapping of specific protective epitopes in antigenic protein molecules which can be used to construct multi-epitope-based antigens for improved vaccine efficacy. Computer models capable of simulating host–vector–pathogen interaction have also been developed, enabling in silico evaluation of candidate universal vaccines prior to testing under field conditions.

The ability of an antigen to induce cross-reactive immunity in the host is essential when considering candidate vaccines for controlling tick infestation by multiple tick species, which is a common occurrence in tropical Africa. However, commercial ATVs have previously shown limited protection against ticks of foreign geographic origins. This suggests the need to identify antigens from local African tick species and to formulate tailor-made vaccines for resident ticks. Such personalized ATVs should also take into account the genotype of the local cattle (hosts) and the epidemiology of TTBDs.

Recent vaccination trials in Uganda support the possibility of formulating ATVs for oral and/or intranasal delivery. This offers the advantage of easy administration, enhanced animal welfare and safety due to reduced stress and risk of contamination or infection at the site of injection. The low immunogenicity and antigen stability after vaccination, however, demand vaccine formulations with selected antigen combinations and optimized immunostimulants. Thus, continuous research and improvement of adjuvants is an essential undertaking for the formulation of efficacious anti-tick vaccines. Another possibility could be the application of oral vaccination in combination with an injectable vaccine. A successful anti-tick vaccination program would be supported by the government and the private sector for sustainability. Indigenous Technical Knowledge (ITK) of livestock keepers on TBDs must also be considered when designing TTBD control strategies. These integrated tick control strategies should overcome common difficulties encountered in the commercialization of tick vaccines.

## Figures and Tables

**Figure 1 vaccines-11-00099-f001:**
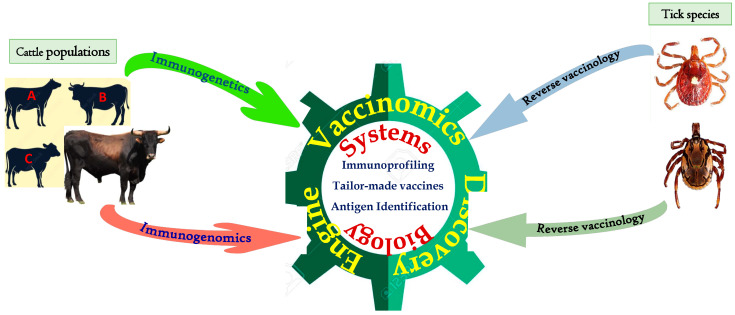
Application of reverse vaccinology to identify anti-tick antigens, and how components of vaccinomics (immunogenetics and immunogenomics) can be integrated with system biology to design cattle population/breed-tailored vaccines against ‘local’ ticks. Letters A, B and C indicate different cattle populations.

**Figure 2 vaccines-11-00099-f002:**
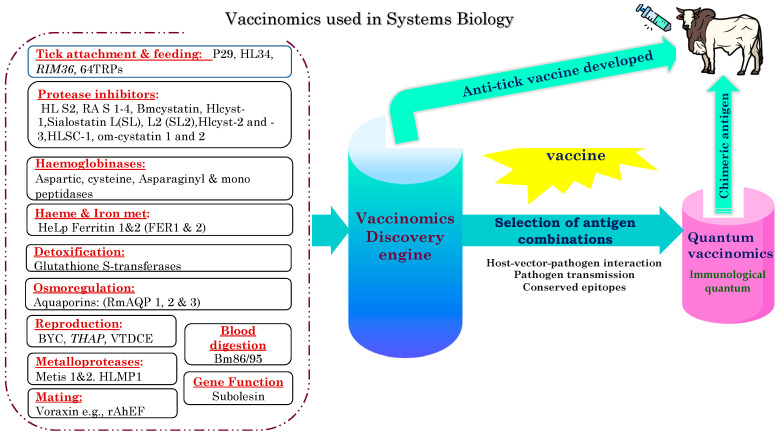
Using vaccinomics in a systems biology approach to identify and characterize protective antigens or select suitable antigen combinations for identification of non-redundant epitopes (immunological quantum) for designing multi-epitope based (chimeric) antigens.

## Data Availability

Not applicable.

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
