# Peer review of "Inspiring Anti-Tick Vaccine Research, Development and Deployment in Tropical Africa for the Control of Cattle Ticks: Review and Insights"

_vaccines, 2022, doi:10.3390/vaccines11010099_

Round 1

Reviewer 1 Report

Kasaija et al., describes the situation, perspective, future of anti-tick vaccination of African grazing ruminants (cattle). The paper contains necessary and useful data for those who works with ticks, their pathogens, and even for the farmers. As I saw the authors tehnd to faciliate this way of prevention, rather than describe its disadventages, costs and questionable success. The paper is written with good English.

-          The two figures are to complex not easy to follow. The authors should think it through what they really tend to underline, to summerize and only give those detailes in Figure(s).

-           The paper should be more concise at least 100 lines could be omitted. At the beginning the reason why  anti-tick vaccination seems beneficial should be given. It is hidden at several places in the text.

The authors should  describe the environment where the vaccinated cattle live. If among large number of free living ruminants (e.g. wildebeest), which feed large amount of ticks vaccination would not work as tick infestation of cattle would be extremely high.

-          How these vaccines work? By antibodies, phagocytic cells?

-          The authors should underline that some agents (which survive in salivary glands, like the tick-borne encephalitis virus) infects the host in 1-2 hours after tick hurted the skin. Vaccination obviously can not work against such infective agents. Shoul be mentioned. Vaccination could affect only agents which enter the host days after the beginning of tick bite (Borrelias).

-          What time the development of immune response against the cement-like material needs? Local inflammation means more active blood flow (rubor) in the histological region which help tick-borne infectious agents to enter the blood stream and it is also speeds up the blood meal process of ticks, left shorter time for the vaccines to act.

Is there described, published scientific work, which gives statistically significant data about decreasing tick bites in a ruminant population by tick-specific immunization. If yes, do not refer only, but give some data. Data about the detailed way of anti tick-immune respnse?

line 547. What does effieciency mean? Ticks detached from the host, or not infested the host at all. How long ticks infected the vaccinated hosts? Amount of proteins? Adjuvant? DNA vaccine? These short data should not be referred (if these data exist at all, should be given).

-          570-578. This section should be part of introduction or somewhere before to show why anti-tick vaccination could be useful and beneficial.

Author Response

Thanks to the reviewer for the effort to review and help to improve this paper.

-     The two figures are to complex not easy to follow. The authors should think it through what they really tend to underline, to summerize and only give those detailes in Figure(s).

  Thanks to the reviewer for the recommendation. The figures have been modified to be easier and clearer to understand.

-     The paper should be more concise at least 100 lines could be omitted. At the beginning the reason why anti-tick vaccination seems beneficial should be given. It is hidden at several places in the text.

      Thanks to the reviewer for the suggestion, the manuscript was revised omitting some sentences and more reasons about why tick vaccines are beneficial have been added in the introduction (lines 59-73).

  • The authors should describe the environment where the vaccinated cattle live. If among large number of free-living ruminants (e.g. wildebeest), which feed large amount of ticks vaccination would not work as tick infestation of cattle would be extremely high.

In response to the reviewer's suggestion, this text was added to the manuscript (lines 495-506): “Exotic cattle breeds (majorly dairy) and their crosses, are more susceptible to TTBDs than indigenous breeds and require heightened protection. Adoption of these commercial breeds in large areas of southwestern and central Uganda demands maintenance of tick-free herds. The animals are mostly kept under a paddocking and small holder dairy production system (zero-grazing), which makes tick control by vaccination possible.

Indigenous cattle in Uganda, like in many parts of sub-Saharan Africa are mainly managed extensively under pastoral and Agro-pastoral systems. The animals are openly grazed and move several kilometers in search of community grazing pastures, which are available in rangelands. These rangelands may also be grazed by wild herbivores such as buffaloes, zebras, rhinos and antelopes which also harbor ticks and provide a source to the livestock. Fortunately, these cattle are relatively resistant to tick infestations and can tolerate ticks for longer periods without intervention. Most veterinarians suggest limited tick control for these local breeds (Kasaija et al., 2020).”

-        How these vaccines work? By antibodies, phagocytic cells?

      Thank you for the question. The information was added in lines 69-73: “The mechanism of protection of tick vaccines is based on the development of antigen-specific antibodies is based on the development of antigen-specific immunized antibodies that immunized antibodies that interact with and affect the function of the antigen in ticks feeding on immunized hosts [6]. This ATVs can affect tick feeding and reproduction controlling tick infestations by reducing tick weight, oviposition and fertility, showing, as a result, a reduction in pathogen prevalence [1, 23].”

      Lines167-175 also, complete this information.

-     The authors should underline that some agents (which survive in salivary glands, like the tick-borne encephalitis virus) infects the host in 1-2 hours after tick hurted the skin. Vaccination obviously can not work against such infective agents. Shoul be mentioned. Vaccination could affect only agents which enter the host days after the beginning of tick bite (Borrelias).

Thank you for the comment. As mentioned in the manuscript, proteins involved in Tick attachment and feeding to repletion have been shown to have an effect on virus infection, such as TBEV. Lines 229-231: R. appendiculatus 64TRPs successfully protected mice against the tick-borne encephalitis virus (TBEV) transmitted by infected I. ricinus ticks to a level comparable to that of a dose of commercial TBEV vaccine [65]. However, this comment was included in the introduction for better understanding (lines 170-173): This may occur within three hours and can last several hours. Whereas pathogens such as the Powassan virus require a short transmission time (15 minutes) and may elude the inflammatory response, most bacterial and protozoan pathogens require several hours of tick feeding before transmission [43, 57].

-     What time the development of immune response against the cement-like material needs? Local inflammation means more active blood flow (rubor) in the histological region which help tick-borne infectious agents to enter the blood stream and it is also speeds up the blood meal process of ticks, left shorter time for the vaccines to act.

Thank you for the question. Responding to the reviewer question this information was added (lines 167-175): The processes leading to acute inflammatory response begin when tissue is first damaged at the tick bite site, allowing a protein-rich exudate containing clotting factors to permeate through the blood vessels to prevent spread of the pathogen. The subsequent migration and degranulation of white blood cells, particularly neutrophils (granulocytes), into the extravascular space of damaged tissue marks the beginning of inflammation. This occurs within three hours and can last several hours. Whereas pathogens such as the Powassan virus require a short transmission time (15 minutes) and could elude the inflammatory response, some vector-borne parasites may not be successfully transmitted [S K Wikel 2018]. For example, the cellular response attracting inflammatory cells to the feeding site of Phlebotomus papatasi is sufficient to block transmission of Leishmania (Valenzuela et al., 2001).

Is there described, published scientific work, which gives statistically significant data about decreasing tick bites in a ruminant population by tick-specific immunization. If yes, do not refer only, but give some data. Data about the detailed way of anti tick-immune respnse?

Thank you for the suggestion. Some efficacy data were added throughout the manuscript, highlighted in yellow

line 547. What does effieciency mean? Ticks detached from the host, or not infested the host at all. How long ticks infected the vaccinated hosts? Amount of proteins? Adjuvant? DNA vaccine? These short data should not be referred (if these data exist at all, should be given).

Reviewer mentioned efficiency by only efficacy is referred in the manuscript considering number of tick larvae, nymphs and adult females completing feeding), oviposition (number of laid eggs) and fertility (number of hatched larvae) calculated as is mentioned in each cited publication. Adjuvant and protein dose depends on the antigen, this review prefers to study the efficacies in screening for useful antigens for tick control in that area of Africa.

      570-578. This section should be part of introduction or somewhere before to show why anti-tick vaccination could be useful and beneficial.

      Thank you for the suggestion. This section was moved to introduction as reviewer recommended (lines 59-65).

Reviewer 2 Report

This is an excellent review of the field. Well done.

Author Response

Thank you very much for revise this manuscript

Reviewer 3 Report

This article is well written and reviews main fields/history of anti-tick vaccine development. I'd like to accept this article with some modification.

1) Title: Title mentions as "in tropical Africa" but the article only mention in case studies in Uganda. I'd like to request the author should describe (review) several previous studies in other African counties.

2) Figures: Difficult to read with many decoration around words, e.g. yellow highlight around protein name in Fig. 2. You have to make more "simple" one.

Author Response

This article is well written and reviews main fields/history of anti-tick vaccine development. I'd like to accept this article with some modification.

Thanks to the reviewer for the effort to review and help to improve this paper.

1)Title: Title mentions as "in tropical Africa" but the article only mention in case studies in Uganda. I'd like to request the author should describe (review) several previous studies in other African counties.

Thank you for the suggestion. To address this comment a new section was included in the manuscript: 6. Anti-tick vaccines in Africa. It includes information from more areas of tropical Africa.

2) Figures: Difficult to read with many decoration around words, e.g. yellow highlight around protein name in Fig. 2. You have to make more "simple" one.

Thank you to the reviewer for the suggestion. The figures have been modified to be easier and clearer to understand.

Round 2

Reviewer 3 Report

Great work!